# Control of cyclic oligoadenylate synthesis in a type III CRISPR system

**Christophe Rouillon, Januka S Athukoralage, Shirley Graham, Sabine Grüschow, Malcolm F White\***

Biomedical Sciences Research Complex, School of Biology, University of St Andrews, St Andrews, United Kingdom

**Abstract** The CRISPR system for prokaryotic adaptive immunity provides RNA-mediated protection from viruses and mobile genetic elements. When viral RNA transcripts are detected, type III systems adopt an activated state that licenses DNA interference and synthesis of cyclic oligoadenylate (cOA). cOA activates nucleases and transcription factors that orchestrate the antiviral response. We demonstrate that cOA synthesis is subject to tight temporal control, commencing on target RNA binding, and is deactivated rapidly as target RNA is cleaved and dissociates. Mismatches in the target RNA are well tolerated and still activate the cyclase domain, except when located close to the 3' end of the target. Phosphorothioate modification reduces target RNA cleavage and stimulates cOA production. The 'RNA shredding' activity originally ascribed to type III systems may thus be a reflection of an exquisite mechanism for control of the Cas10 subunit, rather than a direct antiviral defence.
DOI: https://doi.org/10.7554/eLife.36734.001

## Introduction

CRISPR systems provide adaptive immunity in prokaryotes against mobile genetic elements (MGE). DNA sequences derived from MGE are incorporated into the host genome separated by short direct repeats, forming the CRISPR locus. This is transcribed and processed to generate CRISPR RNAs (crRNA) that are loaded into effector complexes, programming them to detect and subsequently destroy the cognate MGE when the cell next encounters one. Class two effector complexes include Cas9, which has seen wide application in genome engineering. Class one complexes are more intricate, multisubunit systems that use a backbone built around Cas7 subunits which binds the crRNA. This backbone subunit is present in multiple copies and is conserved across the Class one effectors, which are further differentiated into type I, III and IV systems (reviewed in [*Makarova et al., 2015*]). The type I complexes, typified by Cascade (type I-E) from *Escherichia coli*, bind target DNA and then recruit the Cas3 enzyme via an interaction with the Cse1 subunit, resulting in Cas3 mediated DNA degradation (*Hochstrasser et al., 2014*). In contrast, type III complexes bind invading RNA species, activating several different enzymatic activities present within the complex (reviewed in [*Tamulaitis et al., 2017*]). The Cas7-mediated 'backbone cleavage' activity degrades bound target RNA with a characteristic 6-nucleotide spacing (*Staals et al., 2014*; *Tamulaitis et al., 2014*; *Hale et al., 2014*). The HD-nuclease domain of the large Cas10 subunit is responsible for target DNA degradation (*Elmore et al., 2016*; *Estrella et al., 2016*; *Kazlauskiene et al., 2016*; *Jung et al., 2015*; *Han et al., 2017a*). This is activated when target RNA is bound by the complex (*Samai et al., 2015*), providing a mechanism for transcription-dependent DNA targeting (*Deng et al., 2013*) that allows lysogenic phage to persist in the host chromosome (*Goldberg et al., 2014*). Type III CRISPR systems are much more tolerant of mismatches with nucleic acid targets than other types, making viral escape difficult (*Pyenson et al., 2017*; *Manica et al., 2011*; *Manica et al., 2013*).

**\*For correspondence:**
mfw2@st-and.ac.uk

**Competing interests:** The authors declare that no competing interests exist.

**eLife digest** The gene editing tool often known simply as CRISPR has become well known in recent years. Its potential applications are wide ranging, including uses in research, healthcare and agriculture. Yet, the CRISPR system originated in microbes where it helps to protect them from viral infections. Viruses infect by inserting their own genes into a host cell, and – almost like a pair of scissors – the CRISPR system can cut up the virus's DNA to stop infections.

CRISPR experts know the popular form of CRISPR as type II, but there are others. Type III CRISPR is less useful as a genetic tool but does also protect microbes from viruses. In addition to targeting DNA, type III CRISPR targets the related RNA molecules from viruses. When it encounters RNA from a virus, the type III CRISPR produces a small molecule called cyclic oligoadenylate (or cOA for short). The cOA molecule activates enzymes known as non-specific ribonucleases, which can destroy all the RNA in the cell. This defence is a less subtle than that provided type II CRISPR and can also damage the cell by destroying other RNA molecules that the microbes use to survive. As such, proper regulation is essential to prevent the type III system from unnecessarily killing the infected cell.

Rouillon et al. studied the control of the type III CRISPR system from the heat-loving microbe *Sulfolobus solfataricus*, which is found in volcanic springs. This species has been a model for studies of the CRISPR system for many years, in part because its proteins are very stable which makes them easier to work with in the laboratory. The results show that the type III CRISPR makes cOA by combining four molecules of adenosine triphosphate (ATP) into a ring. CRISPR responds immediately to viral RNA in the cell. It also detaches from the RNA as soon as it starts to be destroyed. Rapid activation and silencing of the production cOA ensures that the CRISPR system is tightly controlled.

These findings reveal that cOA production is tightly linked to the abundance of viral RNA, ensuring a proportional and timely response to infection. Using cOA amplifies the cell's response because it allows a single RNA molecule to activate a larger change.

Type III CRISPR systems are widespread in nature, and a better understanding of them could improve the yield of products, like yoghurt, that depend on healthy bacteria; currently viruses cause a lot of economic damage in this industry. Further research in this area could also lead to new antibiotics that over-activate type III CRISPR to destroy bacterial cells.

DOI: https://doi.org/10.7554/eLife.36734.002

Most recently a third activity of type III effector complexes has been described. For some time, it was known that many type III system operons encode a protein such as Csm6 or Csx1 that is not part of the effector complex. These genes have been shown to be important for CRISPR-based immunity in vivo (*Deng et al., 2013*; *Jiang et al., 2016*). The proteins have a CARF (CRISPR-associated Rossmann fold) domain linked to a HEPN (higher eukaryotes and prokaryotes nucleotide binding) domain and have weak ribonuclease activity in vitro (*Niewoehner and Jinek, 2016*; *Sheppard et al., 2016*). CARF domains were predicted to bind some form of cyclic nucleotide, perhaps as an allosteric effector (*Lintner et al., 2011*) but the source of the effector was unknown. This puzzle was solved when it was demonstrated that the cyclase domain of Cas10 can synthesise cyclic oligoadenylate (cOA) molecules from ATP, when activated by target RNA binding (*Kazlauskiene et al., 2017*; *Niewoehner et al., 2017*). cOA species consist of 3 to 6 membered rings of AMP with 3' and 5' linkages and have not been observed in any other biological context. cOA in turn binds to and activates Csm6 and Csx1, enhancing their ribonuclease activity. cOA thus represents a new type of second messenger, generated by type III CRISPR effector complexes, that sculpts the cellular response to invasion by MGE (reviewed in [*Koonin and Makarova, 2018*]).

The crenarchaeaon *Sulfolobus solfataricus* has been a model for studies of the CRISPR-Cas system for many years (reviewed in [*Garrett et al., 2015*]). *S. solfataricus* encodes two type III effector systems: a Cmr (type III-B) complex, which has a unique Cmr7 subunit (*Zhang et al., 2012*), and a Csm (type III-D) complex (*Rouillon et al., 2013*) (*Figure 1*). Both have a Cas7 backbone and a large Cas10 subunit with HD and cyclase domains, but they differ in their structural organisation and nuclease activities (*Zhang et al., 2016*). *S. solfataricus* Cmr has two distinct target RNA cleavage modes and has not been observed to cleave DNA targets (*Zhang et al., 2016*). In contrast the Csm

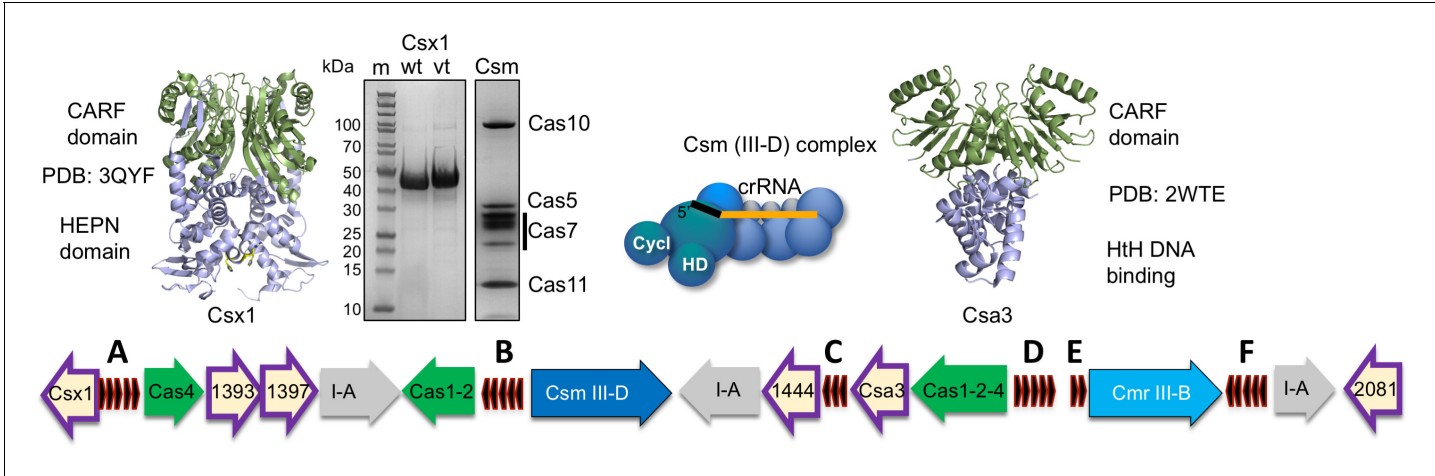

**Figure 1.** The CRISPR locus and cOA signalling proteins in *S. solfataricus*. The six CRISPR loci (**A-F**) are shown in black. There are three type I-A systems (grey), a Csm/III D system (dark blue) and Cmr/III B system (light blue). Adaptation genes are shown in green. Numbers shown correspond to Sso gene numbers; non-Cas genes are omitted. Genes encoding CARF family proteins are outlined in purple. The structures of two CARF proteins, Sso1389 (Csx1) and Sso1445 (Csa3), are shown. The SDS-PAGE gels show purified Csx1 (Sso1389) wild-type and variant H345N, and the Csm complex.
DOI: https://doi.org/10.7554/eLife.36734.003

complex has the canonical Cas7-mediated 'backbone' target RNA cleavage activity and a DNA nuclease activity (*Zhang et al., 2016*). The *S. solfataricus* CRISPR system also has six CRISPR-associated CARF-domain proteins of which one is predicted to be HEPN nucleases (Sso1389), two belong to the Csa3-like transcription factor family (Sso1444 and 1445) and three have indeterminate function (Sso1393, 1397 and 2081) (*Figure 1*). This suggests that *S. solfataricus* can mount a sophisticated and wide-ranging antiviral response through cOA signalling. Large-scale changes in gene expression on viral infection in *S. solfataricus* have been reported (*Quax et al., 2013*), which may be at least partly due to cOA signalling. However, to date, cOA synthesis has not been directly demonstrated in any type III system in the archaea. Recently, a HEPN family Csx1 nuclease from the related organism *S. islandicus* was shown to be activated by mRNA with a 3' polyadenylate tail (*Han et al., 2017b*), which opens another possible mechanism for activation of this antiviral response, independent of cOA signalling. Furthermore, viral infection induces dormancy in *S. islandicus* (*Bautista et al., 2015*), which could be consistent with activation of a cOA-dependent ribonuclease or cOA-dependent changes in transcription.

Here, we demonstrate that the *S. solfataricus* Csm/III-D complex generates cOA in response to target RNA binding, activating a CARF-domain nuclease Csx1 for RNA degradation. We describe a new method to generate short oligoadenosine molecules with a 3'-cyclic phosphate moiety and use this to determine that $cOA_4$ is the relevant activator for Csx1. Activation of cOA synthesis is sensitive to changes at the 3' end of RNA targets, next to Cas10. The deactivation of cOA synthesis is shown to correlate with backbone cleavage towards the centre of bound target RNA, leading to rapid product release.

## Results

### *S. solfataricus* Csm generates cOA in response to target RNA binding

cOA synthesis has so far been observed directly in two bacterial type III systems (*Kazlauskiene et al., 2017*; *Niewoehner et al., 2017*). To study the system in the archaeon *S. solfataricus*, we first purified the Csm complex and the CARF nuclease Csx1 (Sso1389) (*Figure 1*). To determine whether the Csm complex in *S. solfataricus* can synthesise cOA, we incubated Csm with target RNA (A26) in the presence of ATP. Using a radioactively labelled target RNA, we first confirmed that the wild-type Csm complex can bind and cleave the target RNA via 'backbone-mediated' cleavage by the Cas7 subunit (*Figure 2A*), as demonstrated previously (*Zhang et al., 2016*). Three cleavage sites (B1-B3) are clearly visible, and a fourth (B4) is very faint. Targeted mutations that

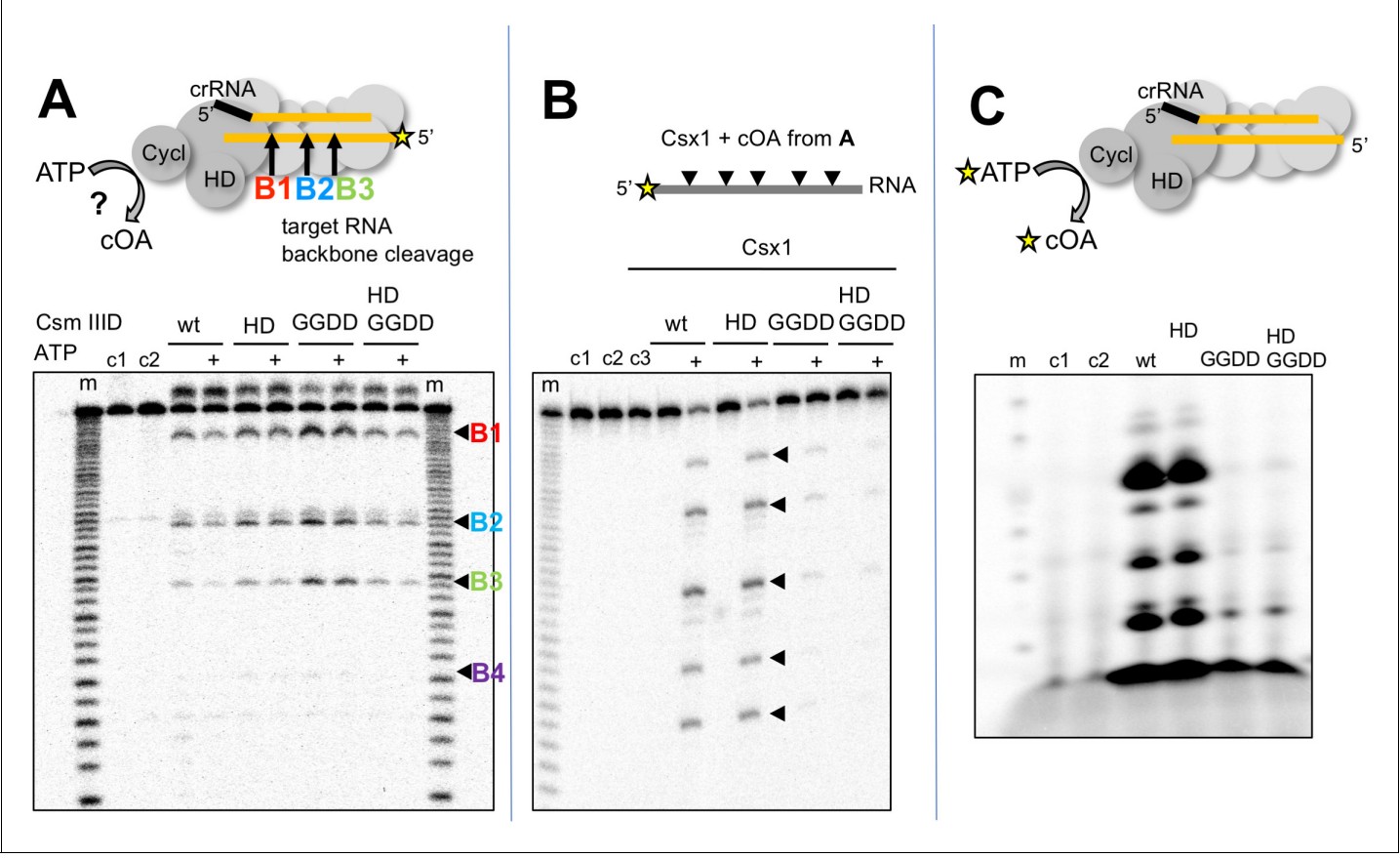

**Figure 2.** *S. solfataricus* Csm synthesises cOA on target RNA binding, activating the Csx1 nuclease. (A) Backbone-mediated cleavage of a labelled target RNA. Three prominent cleavage sites (B1–B3) are indicated. Abrogation of the HD nuclease (HD) or cyclase (GGDD) domain active sites has no effect on this activity. The marker lanes (m) are generated by alkaline hydrolysis of the target RNA. The presence of ATP in the reaction is indicated by a '+' symbol. Control lanes c1 and c2 represent incubation without Csm complex in the presence and absence of ATP, respectively. The diffuse density above the intact RNA is an artefact of electrophoresis. The yellow stars indicate the presence of a radioactive label. (B) Activation of the CARF nuclease Csx1 (Sso1389) by the reaction products of the Csm reaction in A. Csx1 is inactive against the labelled substrate RNA unless activated by the products of the Csm reaction. This activation is dependent on the presence of ATP in the original reaction and an intact cyclase domain. Control lanes c1 and c2 represent incubation of the RNA in buffer at 4 and 50°C, respectively, whereas c3 is a control in presence of Csx1 without added supernatant from Csm reaction. (C) Analysis of the products generated by Csm on addition of target RNA and $\alpha-^{32}$P-ATP. These products are dependent on the presence of an active cyclase domain. They represent a range of linear and cyclic polyadenosines of varying sizes. Control lanes c1 and c2 respectively represent the reaction in absence of Csm or in presence of Csm without target RNA.

DOI: https://doi.org/10.7554/eLife.36734.004

The following figure supplement is available for figure 2:

**Figure supplement 1.** LC-MS analysis of cOA production by S. *solfataricus* Csm.

DOI: https://doi.org/10.7554/eLife.36734.005

abrogate the active site of the HD domain and cyclase domain (denoted HD and GGDD, respectively) did not affect this backbone cleavage mode. The presence of 1 mM ATP in these reactions also had little effect on activity. Next, we performed the same reactions shown in *Figure 2A* but with unlabelled target RNA, phenol-chloroform extracted the products of reactions shown in *Figure 2A* to deproteinise them and added this solution to a reaction containing purified Csx1 (Sso1389) along with a radioactively labelled RNA substrate (A1) (*Figure 2B*). Cleavage of the target RNA by Csx1 at five positions (arrowed) was only observed with products of reactions that had included ATP as well as Csm with a wild-type cyclase domain. The cleavage sites map to C/U dinucleotides for this substrate (see *Table 1*). An active Csm HD domain was not required for stimulation of Csx1. These data suggest that Csx1 is activated by a small molecule that is synthesised by the cyclase domain of Csm when it is incubated with target RNA in the presence of ATP, consistent with

**Table 1.** Oligonucleotides.

Regions complementary to crRNA A26 are italicized and mismatches are shown in bold. crRNA A26 is shown 3' to 5' and the 5' handle is bold. Phosphorothioate linkages are indicated with an asterisk. For substrate A1, Csx1 cleavage sites are in bold.

| crRNA A26 | 3'-GCAACAATTCTTGCTGCAACAATCTTCAACCCATACCAGAAAGUUA |
|---|---|
| Name | Sequence (5'−3') |
| Target RNA A26 | AGGGU*CGUUGUUAAGAACGACGUUGUUAGAAGUUGGGUAUGGUG*GAGA |
| 3'-blunt | AGGGU*CGUUGUUAAGAACGACGUUGUUAGAAGUUGGGUAUGGU* |
| Match | *CGUUGUUAAGAACGACGUUGUUAGAAGUUGGGUAUGGUCUUUCAAU* |
| 5'-blunt | *CGUUGUUAAGAACGACGUUGUUAGAAGUUGGGUAUGGUG*GAGA |
| MM0.5 | AGGGU*CGUUGUUAAGAACGACGUUGUUAGAAGUUGGGUAU**UU**U*GGAGA |
| MM1 | AGGGU*CGUUGUUAAGAACGACGUUGUUAGAAGUUGGGU**CG**G*GUGGAGA |
| MM1a | AGGGU*CGUUGUUAAGAACGACGUUGUUAGAAGUUGGGUA**G**GG*UGGAGA |
| MM1b | AGGGU*CGUUGUUAAGAACGACGUUGUUAGAAGUUGGGU**C**UG*GUGGAGA |
| MM1.5 | AGGGU*CGUUGUUAAGAACGACGUUGUUAGAAG**CC**GGG*UAUGGUGGAGA |
| MM2 | AGGGU*CGUUGUUAAGAACGACGUUGU**CG**GAAGUUGGG*UAUGGUGGAGA |
| MM2.5 | AGGGU*CGUUGUUAAGAACGACGUU**CA**UAGAAGUUGGGUAUGG*UG*GAGA |
| MM4 | AGGGU*CGUUGUUAA**AG**ACGACGUUGUUAGAAGUUGGGUAUGGUG*GAGA |
| P-thioate A26 | AGGGU*CGUUGUUAAGAACGACGUUGU\*U\*A\*GAAGUUGGGU\*A\*U\*GGUGGAGA |
| Substrate (A1) | AGGGUAUUAUUUGUUUGUUU**CUU****CU**AAA**CU**AUAAG**CU**AGUU**CU**GGAGA |
| A3 (MazF) | AAAACAUCAG |
| A4 | AAAAACAUCAG |
| A5 | AAAAAACAUCAG |
| A6 | AAAAAAACAUCAG |
| Trap DNA | GTCGTTCTTAACAACGACCCT |
| Target RNA B2 product | AGGGU*CGUUGUUAAGAACGACGUUGU*U |

DOI: https://doi.org/10.7554/eLife.36734.006

an active cOA-signalling pathway in *S. solfataricus*. By carrying out the reaction shown in *Figure 2A* in the presence of $\alpha$-$^{32}$P-ATP, followed by separation of reaction products by denaturing gel electrophoresis, we observed synthesis of small radioactive oligomers, dependent on an active Csm cyclase domain, consistent with cOA synthesis by the Csm complex (*Figure 2C*). Liquid chromatography coupled with mass spectrometry (LC-MS) was used to confirm that the major product corresponded to cOA$_4$, with much lower amounts of other cyclic molecules present (*Figure 2—figure supplement 1*).

## Linear analogues of cOA generated using the MazF toxin confirm cOA$_4$ as the relevant activator

Although cOA$_4$ is the main product generated by the Csm cyclase domain, it is possible that Csx1 is stimulated by a minor product. For example, *S. thermophilus* Csm mainly generates cOA$_3$ and cOA$_4$ but very little of the cOA$_6$ that is the activator of the cognate Csm6 (*Kazlauskiene et al., 2017*). It is difficult to generate large amounts of cOA of defined ring size by enzymatic means; the type III systems are complex ribonucleoprotein machines, and each generates a mixture of cOA molecules with different ring sizes, often in non-ideal proportions. Linear oligoadenylates can activate CARF-domain nucleases, an effect stimulated by the presence of a cyclic 2',3'-phosphate terminus (*Niewoehner et al., 2017*). However, standard phosphoramidite synthesis is problematic for small oligonucleotides below six nucleotides in length. In order to generate large quantities of defined signalling species, we have devised a convenient method for the generation of linear polyadenylates with a 2',3' cyclic phosphate – a species that has an identical mass to the corresponding cOA. To do this, we took advantage of the specificity of the MazF toxin of *E. coli*, which cuts RNA 5' to an ACA recognition sequence, leaving a 2',3'-cyclic phosphate (*Zhang et al., 2003*). MazF can be expressed

at high levels in *E. coli* in complex with the antitoxin MazE and is activated by proteolysis (*Park et al., 2012*). By feeding recombinant MazF with oligoribonucleotides of sequence 5'-(A)$_n$-ACAUCAG, we could generate large quantities of linear analogues of cOAs A$_3$ >P, A$_4$ >P, A$_5$ >P and A$_6$ >P (where '>P' denotes a 2',3' cyclic phosphate moiety) as desired (*Figure 3A*).

To test the utility of this approach, we incubated the Csx1 nuclease Sso1389 with a labelled target RNA substrate in the presence of MazF-derived A$_3$ >P, A$_4$ >P, A$_5$ >P and A$_6$ >P oligoadenylates (*Figure 3B*). We observed clear activation of the Csx1 ribonuclease activity by A$_4$ >P, comparable to the activation by the authentic cOA mixture generated by the *S. solfataricus* Csm effector, demonstrating that cOA$_4$ is the relevant activator for this enzyme.

## cOA synthesis is coupled to target RNA cleavage

To investigate the relationship between target RNA cleavage and cOA synthesis, we used a 205 nt RNA transcript containing the target site for crRNA A26 (*Figure 4*). The RNA transcript was labelled during in vitro transcription with α-$^{32}$P-ATP, allowing detection of any product containing an

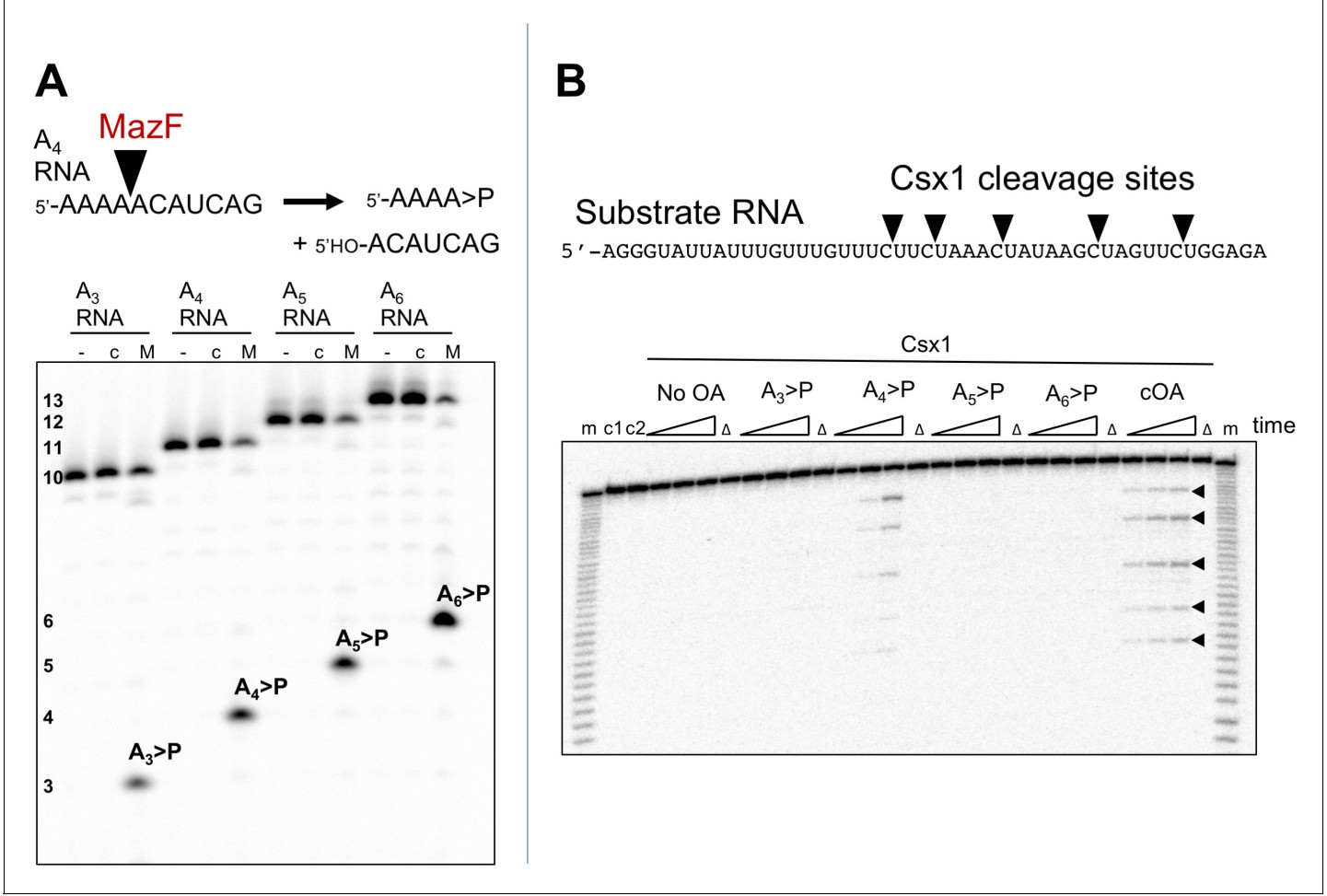

**Figure 3.** The Csx1 nuclease is activated by linear cOA analogues generated by MazF. (A) Synthesis of linear A$_3$ – A$_6$ oligoadenylate molecules with cyclic 3'-phosphate termini by MazF. MazF cleaves on the 5' side of an ACA recognition sequence. For each reaction set, lane '-' is RNA substrate; 'c' is the incubation of the RNA in reaction buffer without added MazF in presence of trypsin; 'M' is RNA incubated with active MazF for 30 min. (B) Purified linear oligoadenylate species were incubated with Csx1 nuclease along with a $^{32}$P-labelled RNA substrate (A1) for 5, 15, 30 min. No Csx1 nuclease activity was detected in the absence of oligoadenylate. When reactions were supplemented by the products of the reaction with Csm, ATP and target RNA (visualised in *Figure 2C*) the nuclease activity of Csx1 was strongly activated. Screening of linear A$_3$ – A$_6$ >P demonstrated that A$_4$ >P is the cognate activator for Csx1. Lane m shows an alkaline hydrolysis ladder of the substrate RNA; lanes marked 'Δ' have the inactive variant of Csx1, H345N, instead of the wild-type enzyme, incubated for 30 min at 50°C; c1 and c2 represent RNA incubated without Csx1 for 30 min at 4 and 50°C, respectively.
DOI: https://doi.org/10.7554/eLife.36734.007

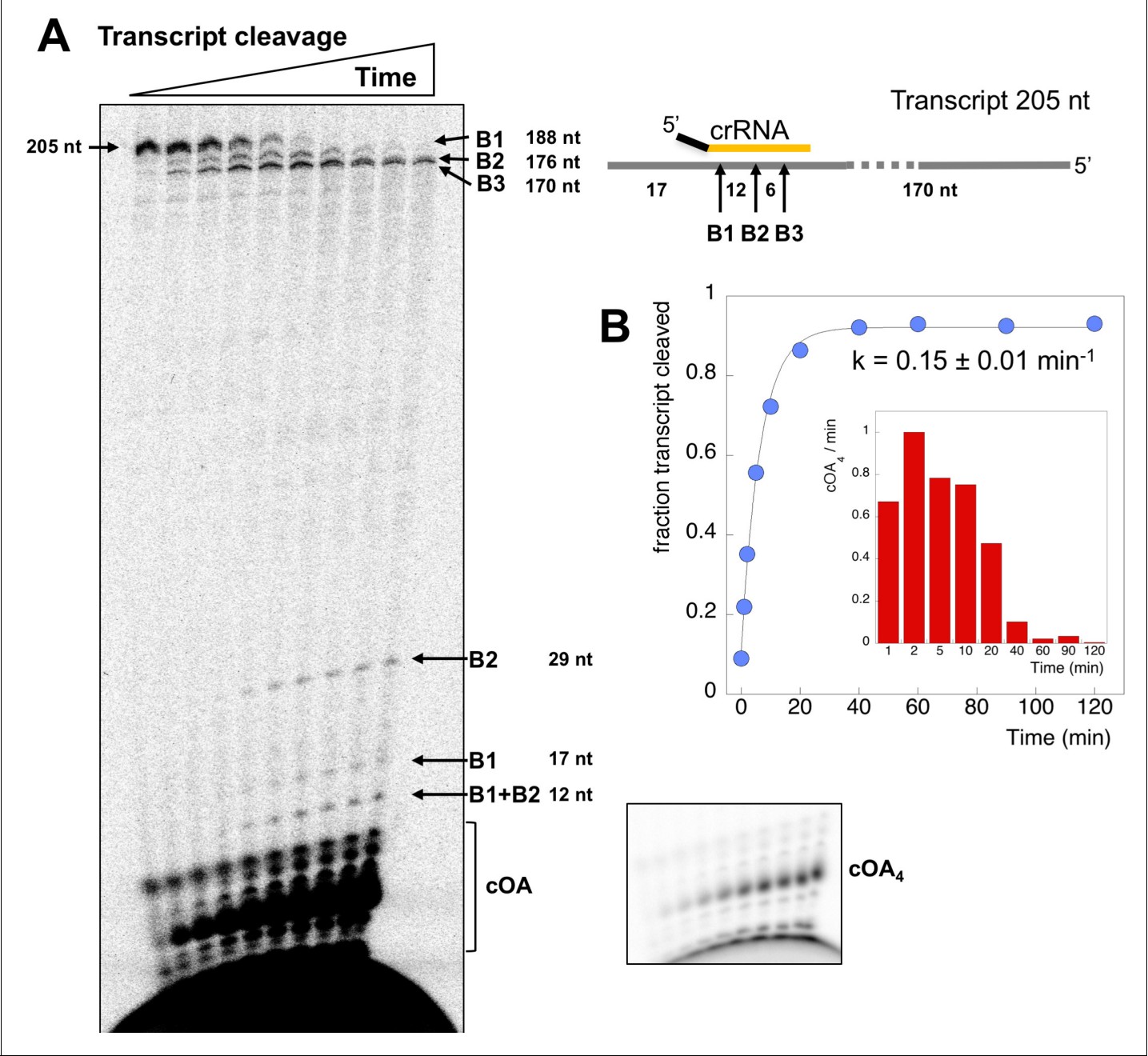

**Figure 4.** Csm transcript cleavage and cOA production. (**A**) Denaturing gel electrophoresis showing cleavage of a 205 nt RNA tanscript target by Csm over time, and cOA synthesis. Products arising from cleavage at sites B1-B3 are indicated. Lanes 1–10 represent time points of 0, 1, 2, 5, 10, 20, 40, 60, 90, 120 min. (**B**) Quantification of the fraction of cleaved transcript against time, fitted to an exponential equation, yielded a rate of 0.15 min⁻¹. A bar chart showing the change in cOA$_4$ production per minute for each time period (normalised to the value at 2 min) is shown as an insert.

DOI: https://doi.org/10.7554/eLife.36734.008

The following source data is available for figure 4:

**Source data 1.** Raw data for the kinetic analysis and cOA production presented in *Figure 4B*.

DOI: https://doi.org/10.7554/eLife.36734.009

adenosine. We showed previously that Csm degrades target RNA oligonucleotides by backbone cleavage at up to four positions, B1-B4 (28). Under single turnover conditions (Csm 5 µM, transcript 5 nM), Csm-mediated backbone cleavage at sites B1-B3 of the transcript target sequence could be observed both by the processing of the large RNA species from 205 down to 170 nt, and by the appearance of small RNA products corresponding to cleavage at sites B1 and B2 further down the gel (*Figure 4A*). Since the six nucleotides between sites B2 and B3 lack an adenosine, this product was not visible. We quantified the fractional cleavage of the transcript over time and fitted it to an exponential equation, yielding a rate $k_c = 0.15 \pm 0.01$ min$^{-1}$ (*Figure 4B*). More detailed kinetics are described using synthetic oligonucleotide substrates, below.

In the same reaction, we monitored cOA production by including $\alpha$-$^{32}$P-ATP (5 nM) along with cold ATP (0.5 mM). cOA synthesis was observed from early time points and plateaued over the time course. No more than 10% of the total amount of ATP was consumed in the course of the reaction. The change in cOA$_4$ production over time was calculated by quantifying the amount of cOA$_4$ produced per minute during each time interval and is shown as a bar chart in *Figure 4B*. As the target site in the transcript RNA becomes fully cleaved between 20 and 40 min, cOA$_4$ production shuts down.

## Target RNA requirements for activation of cOA synthesis

Induction of cOA synthesis in response to viral infection induces an anti-viral state with an altered gene expression landscape and activation of non-specific ribonucleases. It follows that the synthesis of cOA should be under tight control. Activation of the cyclase domain of the Csm complex occurs on target RNA binding (*Figure 2*), which presumably initiates a conformational change that activates the cyclase domain. To investigate this further, we tested the activation of cOA synthesis using a variety of oligonucleotide target RNA molecules (*Figure 5*). Viral-derived target RNA is naturally mismatched with the CRISPR-derived 5' handle of the crRNA. When the target RNA is complementary to the 5' handle of the crRNA sequence, or when the 5 nt mismatched 3'-end of target RNA is missing, backbone cleavage still occurs but the cyclase domain is not activated (*Figure 5—figure supplement 1*). This confirms previous observations with *S. thermophilus* Csm (*Kazlauskiene et al., 2017*). We next tested the effect of mismatches by introducing pairs of mismatches at a variety of positions along the target RNA (*Figure 5*). All mismatched targets were still cleaved by Csm, in keeping with previous observations (*Staals et al., 2014*; *Kazlauskiene et al., 2016*). Target MM1 and MM2, with two mismatches spanning cleavage site B1 and B2 respectively, abolished backbone cleavage at the mutated position, but cleavage at other sites was still observed. Single mismatches flanking site B1 reduced but did not abolish backbone cleavage at site B1 (MM1a and 1b), whilst the double mismatch adjacent to site B1 (MM0.5) reduced cleavage at that site very significantly, suggesting local perturbation. None of the double mismatches introduced 5' to site B1 on the target RNA reduced cOA synthesis, showing that they behave very similarly to fully cognate target RNA. In contrast, a double mismatch positioned two nt (MM0.5) or four nt (MM1) from the 3' end of the matching target RNA resulted in complete abolition of cOA synthesis. Together, these data suggest that Csm is particularly sensitive to disruptions of targets at the 3' end nearest the Cas10 subunit, whether by truncation, base-pairing with the crRNA 5'-handle, or mismatch (discussed later).

## Kinetic analysis of target RNA cleavage

Two possibilities for deactivation of the cyclase activity are when target RNA is cleaved, or when cleaved target RNA is released from the Csm complex. To measure the rates of target RNA cleavage and product release under pseudo-single turnover conditions we incubated an excess of Csm with an end-labelled target RNA oligonucleotide to follow target cleavage as a function of time. Three backbone cleavage sites (B1-B3) were clearly visible on gels (*Figure 6A*) (*Zhang et al., 2016*). We quantified the rates of cleavage at these three sites in triplicate experiments (*Figure 6B*), normalised to aid comparison – this normalisation does not change the rates measured. Site B1, which is the closest site to the Cas10 subunit, was cleaved with a rate constant $k_{B1}$ of 0.19 min$^{-1}$ under these conditions. The next backbone cleavage site, B2, was cut about half as fast ($k_{B2}$ = 0.077 min$^{-1}$). The third site, distal to Cas10, was cleaved at the slowest rate ($k_{B3}$ = 0.004 min$^{-1}$). The rate of cleavage at sites B1 and B2 thus corresponds very well with that observed for the transcript substrate (*Figure 4*). Faster target RNA cleavage at sites closer to the Cas10 domain has been observed

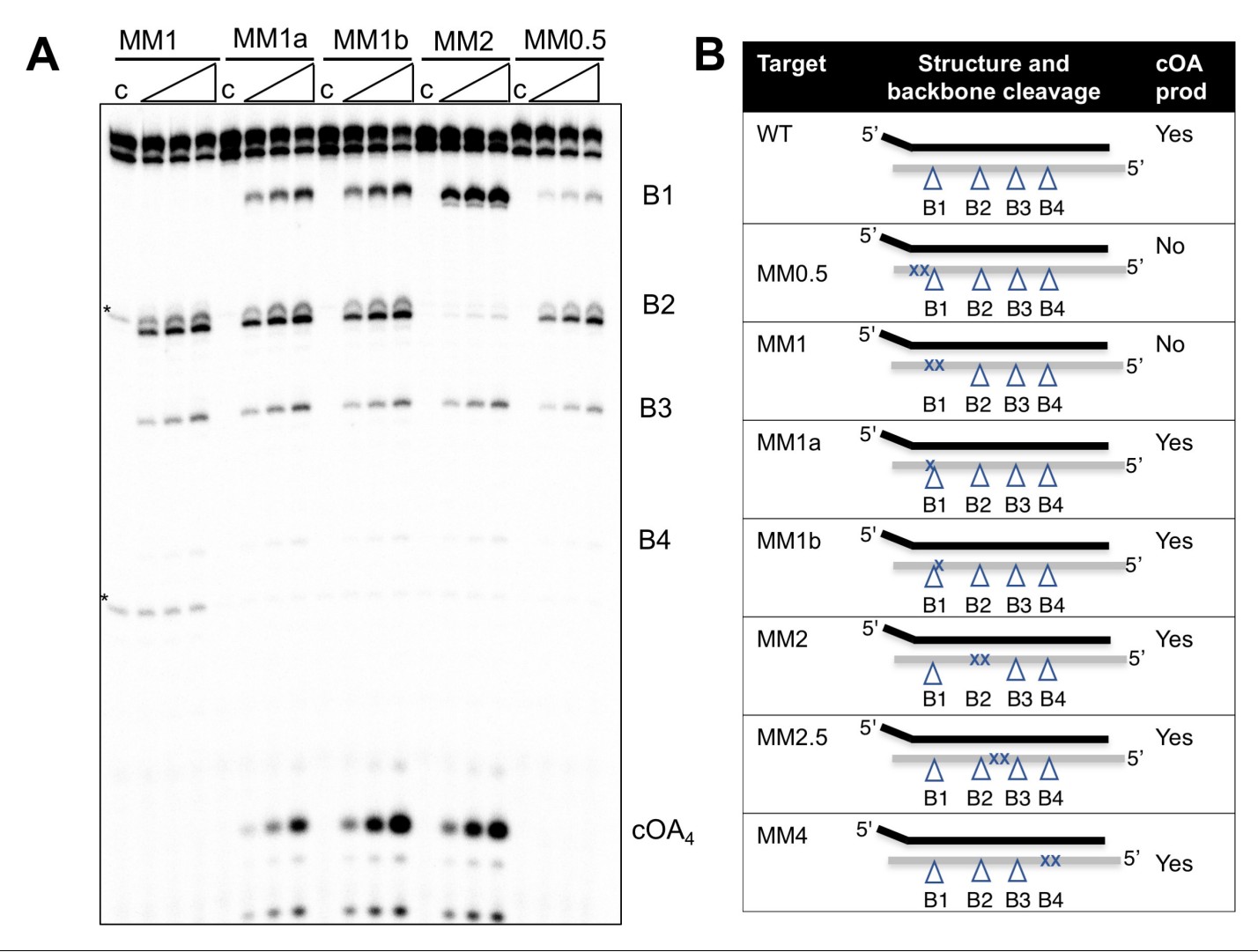

**Figure 5.** Effect of mismatched targets on cOA production. Oligonucleotides containing mismatches ranging along the length of the target RNA were used to test both backbone cleavage and cOA synthesis activity by Csm. Backbone cleavage is abolished when two mismatches flank the cleavage site, but is otherwise active. cOA synthesis is not observed for substrates MM0.5 and MM1, which introduce two consecutive mismatches close to the 3' end of the target RNA. Bands marked with an asterisk are artefactual bands present in the RNA substrate. For all targets, lane c shows the reaction at the zero time point and the three further time points correspond to incubation for 5, 10 and 20 min. Further substrates are shown in *Figure 5—figure supplement 1*. (B) Table summarising backbone cleavage and cOA production for the mismatch substrates studied in *Figure 5A* and *Figure 5—figure supplement 1*. Backbone cleavage sites are indicated with a triangle and mismatches with an 'X'. The target RNA is represented by the grey line.
DOI: https://doi.org/10.7554/eLife.36734.010

The following figure supplement is available for figure 5:

**Figure supplement 1.** Backbone cleavage and cOA production for target RNAs with altered 5' and 3' ends.
DOI: https://doi.org/10.7554/eLife.36734.011

qualitatively for other type III systems (*Staals et al., 2014*; *Tamulaitis et al., 2014*), and may be a general phenomenon.

## Kinetic analysis of cOA synthesis

Having defined the kinetics of target RNA cleavage, we investigated the related cOA synthesis activity of Csm by feeding the reaction with $\alpha$-[32]P-ATP to monitor the synthesis of cOA species. These experiments were carried out under the exact same conditions as those used to determine RNA cleavage (in fact they can be run simultaneously with labelled oligonucleotide target RNA and $\alpha$-[32]P-

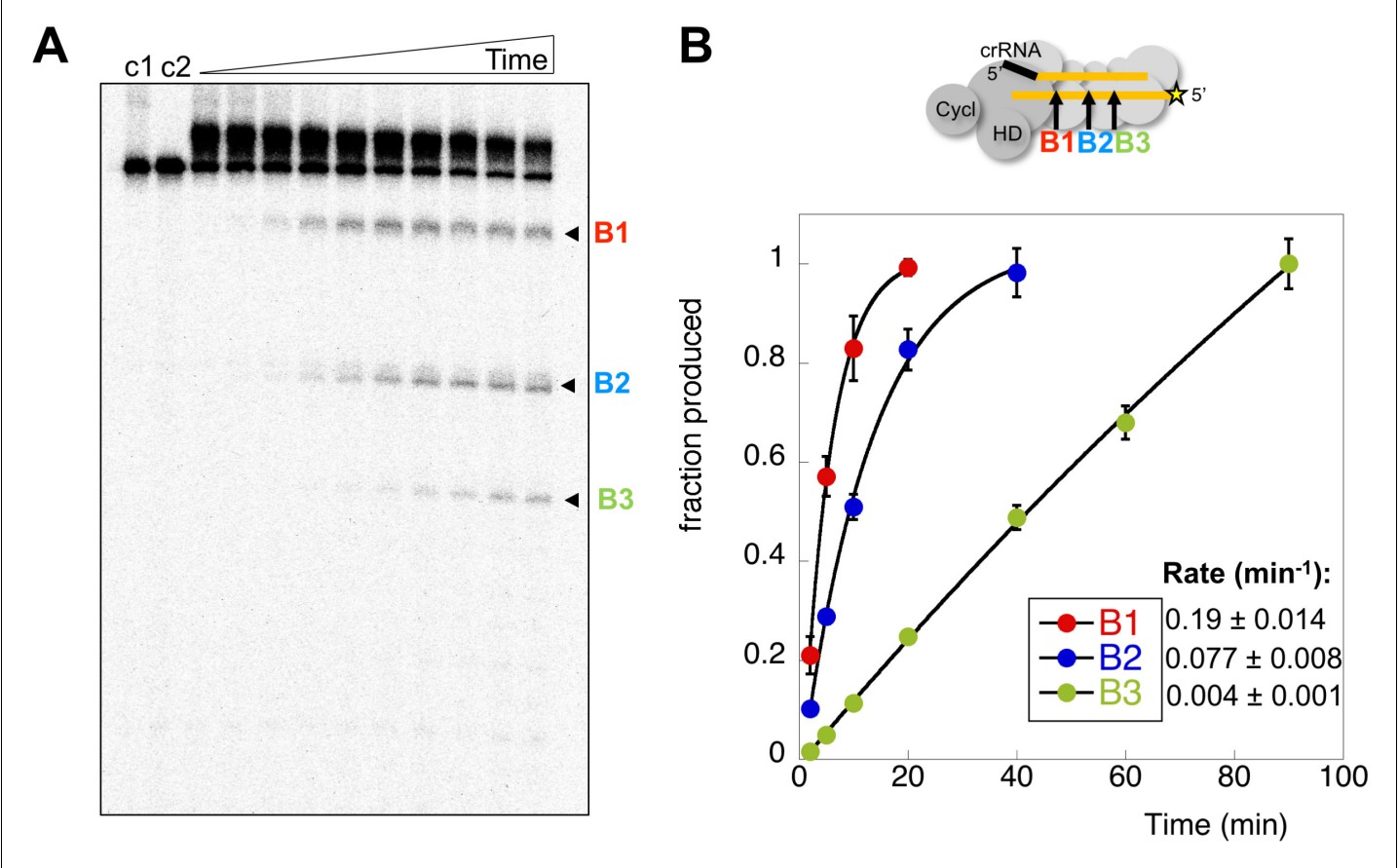

**Figure 6.** Rates of target RNA cleavage at each position. (**A**) Backbone cleavage of target RNA over time (0,1, 2, 5, 10, 20, 40, 60, 90, 120 min). Controls c1 and c2 represent target RNA incubated for 120 min in the absence of Csm at 4 and 70°C, respectively. (**B**) The rates of target RNA backbone cleavage at sites B1-B3 were determined. Data points represent means of 4 experiments with standard deviation shown. The star represents the radioactive label.

DOI: https://doi.org/10.7554/eLife.36734.012

The following source data is available for figure 6:

**Source data 1.** Raw data for the kinetic analysis presented in *Figure 6B*.
DOI: https://doi.org/10.7554/eLife.36734.013

ATP). The amount of $cOA_4$ generated per minute during each interval following the initiation of the reaction was quantified in triplicate and is shown in *Figure 7A*. The rate of $cOA_4$ synthesis peaked at 10 min and rapidly fell away, suggesting that the cyclase domain is rapidly activated on target RNA binding, and quickly deactivated after target RNA cleavage. To investigate this further, we monitored $cOA_4$ synthesis by spiking radioactive ATP into a reaction containing Csm, target RNA and 0.5 mM cold ATP at three different time points (0, 20 and 40 min) after initiation of the reaction and followed $cOA_4$ synthesis for a further 40 min (*Figure 7B*). Under single turnover conditions with 25 nM target RNA, we observed very low amounts of $cOA_4$ production after 20 min of reaction. When the target RNA was increased ten-fold, this allowed cOA production to continue for a longer period. Thus, cOA synthesis is closely linked to target RNA availability, and is rapidly deactivated once target RNA is cleaved.

## Phosphorothioate modification slows target RNA degradation and enhances cOA production

Oligonucleotides with phosphorothioate (P-thioate) bonds, where a non-bridging oxygen of the phosphate linkage is replaced by sulfur, show enhanced resistance to a variety of nucleases. We therefore explored the effect of introduction of three consecutive phosphorothioate linkages

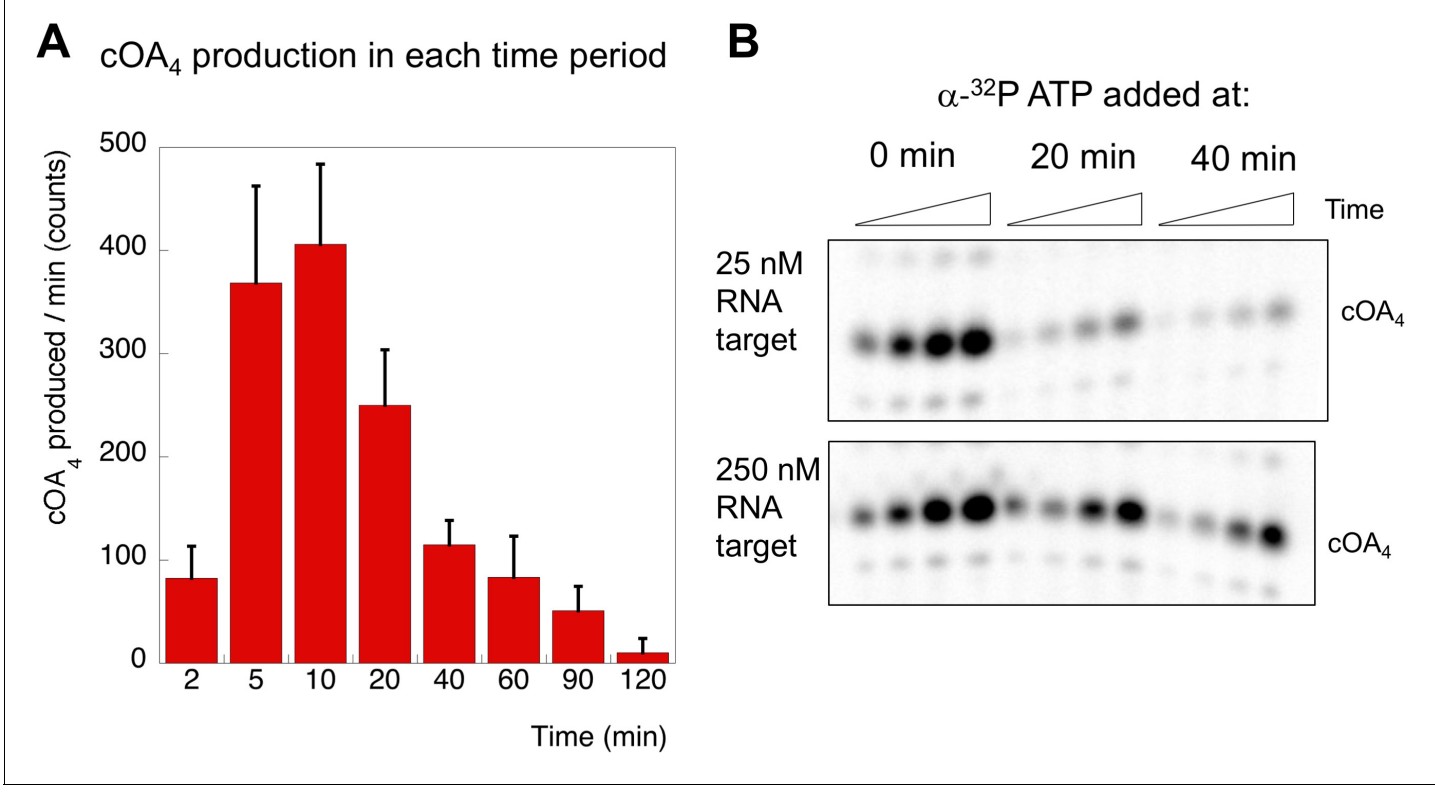

**Figure 7.** cOA production as a function of time and target RNA concentration. (A) $cOA_4$ synthesis was quantified using radioactive ATP under the assay conditions described for *Figure 6* and plotted as $cOA_4$ synthesised per min (densitometric counts) during the period of each reaction time slot. This was carried out in triplicate: means and standard deviations are shown. (B) In this experiment, $\alpha$-$^{32}$P-ATP was added to reactions at 0, 20 or 40 min and the reaction followed for a further 40 min (time points 5, 10, 20 and 40 min). Under standard single turnover conditions (25 nM target RNA), cOA synthesis was largely complete after 20 min. When target RNA was increased ten-fold, cOA synthesis persisted for a longer period.

DOI: https://doi.org/10.7554/eLife.36734.014

The following source data is available for figure 7:

**Source data 1.** Raw data for the kinetic analysis presented in *Figure 7A*.

DOI: https://doi.org/10.7554/eLife.36734.015

centred on cleavage sites B1 and B2 (*Table 1*). The P-thioate target RNA was still cleaved by Csm, but the rate was significantly slower (*Figure 8*). In particular, a defect in cleavage at site B2 was apparent. By adding radioactive $\alpha$-ATP to the reaction, we were able to monitor $cOA_4$ simultaneously. This revealed that $cOA_4$ production was strongly enhanced in the reaction with P-thioate target RNA, consistent with a requirement for cleavage and/or dissociation of cleaved target RNA to switch off the activity of the Cyclase domain.

## Kinetic analysis of RNA product release

In the cell, target RNA cleavage by Csm may be rate-limited by the rate of product release, rather than the chemical step of a cleavage reaction (*Kazlauskiene et al., 2016*). Conceivably, cleaved target RNA that remained bound to Csm could continue to activate the cyclase domain. We therefore quantified the rate of dissociation of cleaved target RNA by carrying out a cleavage reaction over time and running the products in a native gel. RNA remaining bound to Csm was held up in the wells whilst released RNA migrated through the gel. Dissociated, radioactively labelled target RNA cleavage products were trapped by the addition of a cold DNA oligonucleotide of complementary sequence to generate a RNA:DNA heteroduplex that migrated at a defined position in the gel (*Figure 9A*). The DNA trap is complementary to the target RNA from the 5' end to roughly the site of B3 cleavage (*Table 1*). The dissociated RNA product built up over the time course of the reaction and was quantified in triplicate experiments. The rate of cleaved target RNA release was determined

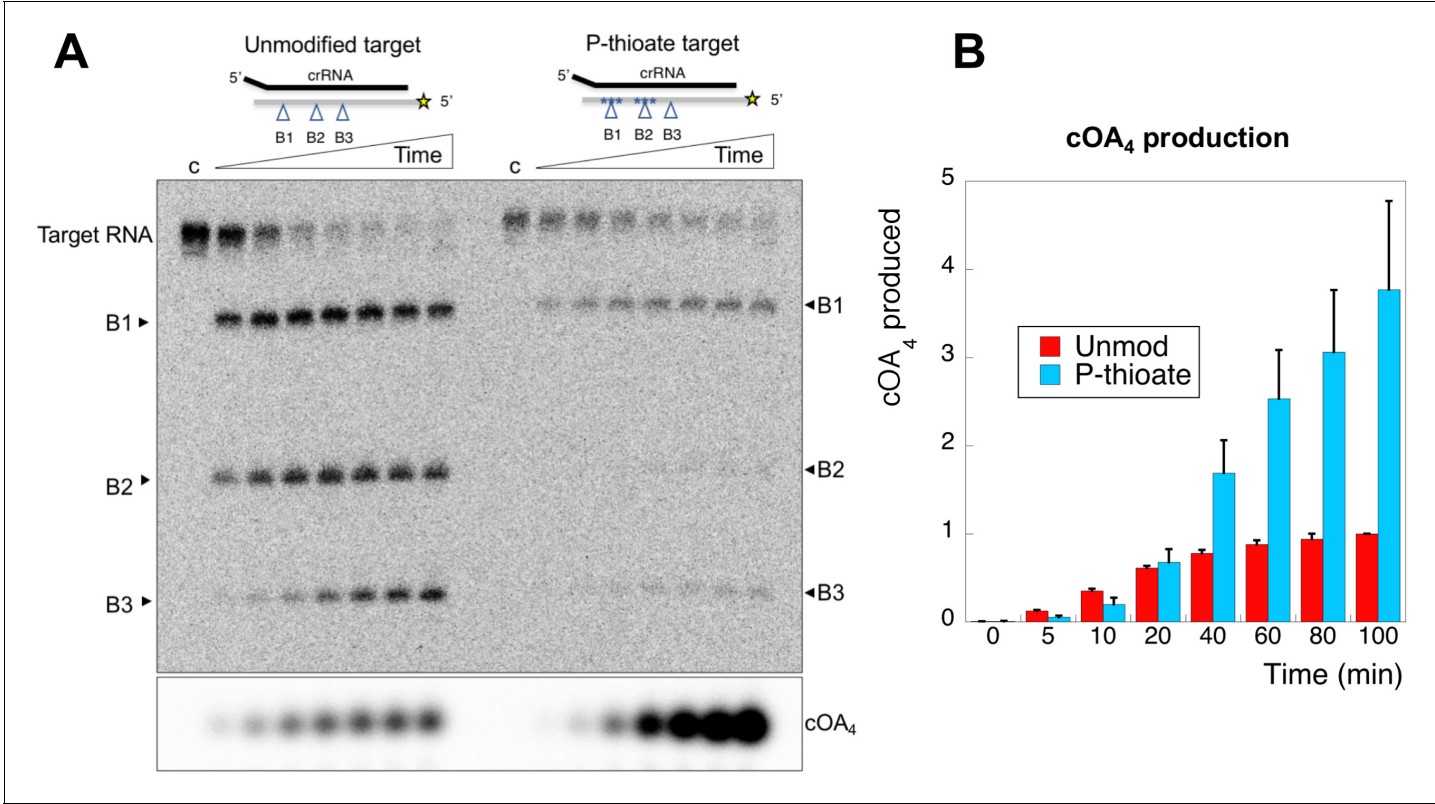

**Figure 8.** An oligonucleotide with phosphorothioate linkages reduces target RNA cleavage and enhances cOA production. (**A**) Comparison of target RNA cleavage for unmodified target RNA (A26) and the same sequence with two sets of three phosphorothioate bonds at sites B1 and B2 (indicated by *). Target RNA cleavage was slowed significantly, with a particular reduction in cleavage at site B2. The synthesis of $cOA_4$, on the other hand, was strongly stimulated over the course of the reaction. Time points are 0 (denoted 'c'), 5, 10, 20, 40, 60, 80, 100 min. (**B**) Quantification of $cOA_4$ production, showing four-fold higher levels of $cOA_4$ after 100 min when the P-thioate modified target RNA was used. Values are the means of triplicate experiments, with standard deviation shown, and are normalised to the signal from the unmodified target RNA at 100 min.

DOI: https://doi.org/10.7554/eLife.36734.016

The following source data is available for figure 8:

**Source data 1.** Raw data for the cOA production data, quantified in triplicate.

DOI: https://doi.org/10.7554/eLife.36734.017

as $k_{off} = 0.066 \pm 0.003$ min$^{-1}$ (**Figure 9B**), similar to the rate of cleavage of site B2 in the centre of the target RNA ($k_{B2} = 0.077$ min$^{-1}$), significantly faster than the cleavage at site B3. The site of cleavage of the main dissociated product was confirmed as B2 using a synthetic RNA marker of that size (**Figure 9—figure supplement 1**). Thus, the clearance of target RNA from the Csm complex may be governed by the rate of backbone cleavage rather than product dissociation. Furthermore, cleavage up to the centre of the target RNA (site B2) appears sufficient to allow product dissociation. We also analysed the rate of synthesis of $cOA_4$ over time (as shown in **Figure 7**), setting the rate at 10 min equal to one and normalising the data. The data fit a simple exponential with a rate constant $k_{obs} = 0.047 \pm 0.008$ min$^{-1}$.

In conclusion, the rates of target RNA cleavage at site B2, RNA product dissociation and inactivation of cOA synthesis were very similar, within experimental error. When cleavage at site B2 is perturbed by phosphorothioate modification, cOA production is enhanced significantly. These observations are consistent with activation of the cyclase domain quickly upon target RNA binding, followed by deactivation as target RNA is cleaved and dissociates from the Csm complex.

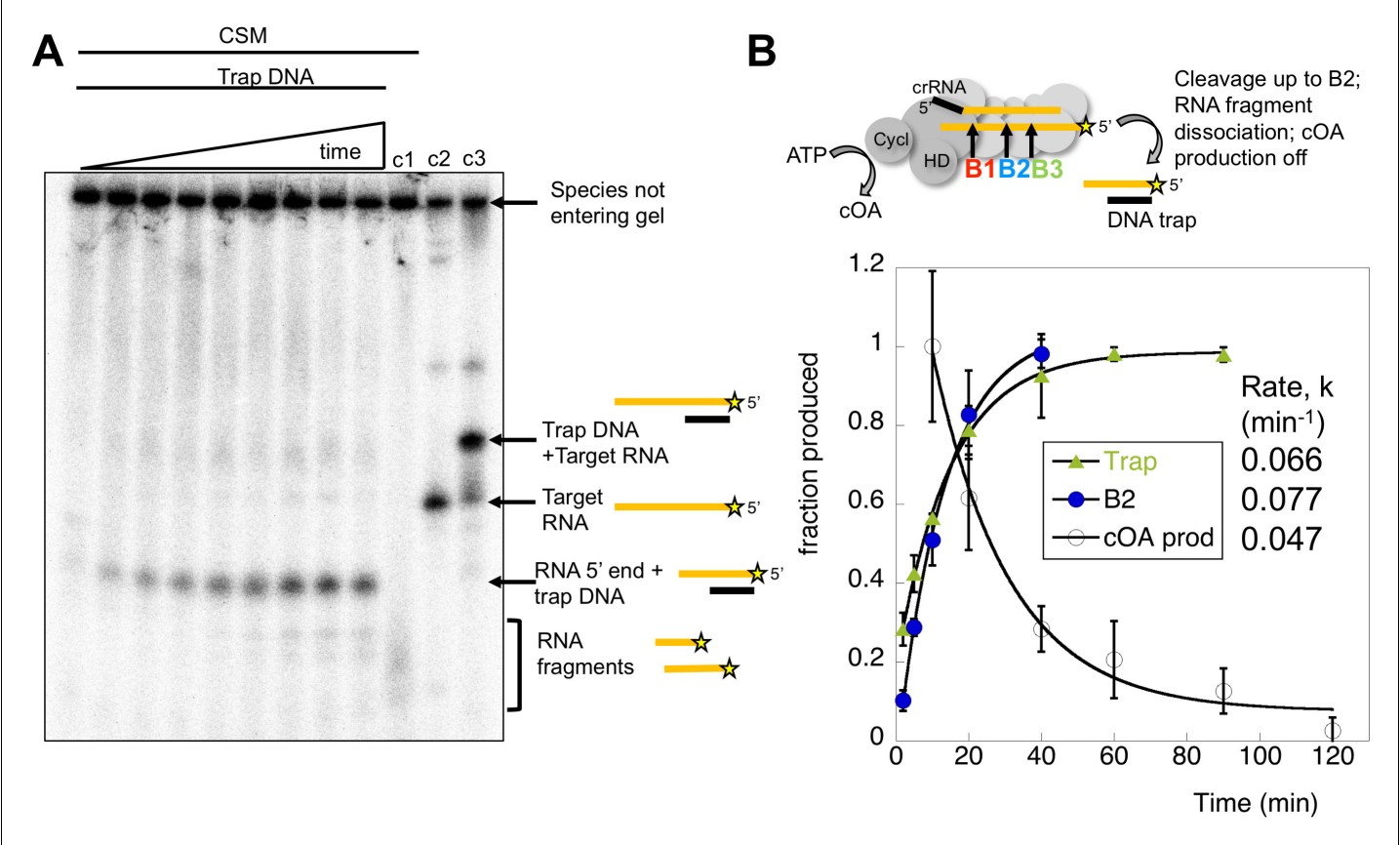

**Figure 9.** Kinetic analysis of the inactivation of cOA synthesis. (**A**) Native polyacrylamide gel following the reaction products generated by Csm bound to radioactively labelled target RNA. A time course of cleavage is shown in the first nine lanes (0, 2, 5, 10, 20, 40, 60, 90, 120 min), where trapping DNA complementary to the 5' end of the target RNA was added at the end of the reaction. Only species released from the Csm complex enter the gel. A time dependent increase in a species corresponding to a trapped RNA:DNA heteroduplex was observed. Lane c1 shows the reaction run for 2 hr in the absence of trapping DNA – a smear of ssRNA reaction products were observed. Lane c2 shows the position of full length target RNA, and lane c3 shows the full length target RNA bound to the trapping DNA. *Figure 9—figure supplement 1* shows the position of a control RNA corresponding to the 5' end of the B2 cleavage product. (**B**) The time-dependent accumulation of trapped RNA was quantified in triplicate, normalised and plotted. Means plus standard deviation are shown, yielding a rate $k_{off} = 0.066$ min$^{-1}$. The progress of cleavage at site B2 (0.077 min$^{-1}$) is included for comparison. We also quantified the rate of cOA synthesis in triplicate using target RNA and normalising the rate at 10 min to the value 1.0. This fitted to an exponential with a rate constant $k_{obs} = 0.047$ min$^{-1}$, suggesting that product release is correlated with a rapid inactivation of cyclase activity.
DOI: https://doi.org/10.7554/eLife.36734.018

The following source data and figure supplement are available for figure 9:

**Source data 1.** Raw data for the kinetic analysis presented in *Figure 9B*.
DOI: https://doi.org/10.7554/eLife.36734.020

**Figure supplement 1.** Target RNA cleaved at B2 is the main released product.
DOI: https://doi.org/10.7554/eLife.36734.019

## Discussion

### Synthesis of cyclic oligoadenylates by type III systems

We have demonstrated that the type III-D Csm complex from *S. solfataricus* generates cOA on binding target RNA. Given the preponderance of CARF-family proteins associated with type III systems and the conservation of the cyclase domain, this is likely to be a general phenomenon. The major product of cOA synthesis by this complex is cOA$_4$, with much smaller amounts of other cyclic and linear oligoadenylates observed. For the *S. solfataricus* system, the generation of cOA$_4$ fits with the binding pocket modelled for the CARF domain of the transcription factor Csa3 (*Lintner et al., 2011*), and with the observation that OA$_4$ activates the CARF nuclease Csx1 in the related organism

*S. islandicus* (*Han et al., 2017b*). *Enterococcus italicus* Csm generates mostly cOA$_6$, and this molecule is the effector for the corresponding Csm6 enzyme (*Niewoehner et al., 2017*). In contrast, the major product generated by the *S. thermophilus* enzyme is cOA$_3$, despite the fact that the corresponding Csm6 protein is activated by cOA$_6$ (*Kazlauskiene et al., 2017*). Given that CARF domains are formed by dimeric subunits, the corresponding activators are likely to have a 2-fold axis of symmetry, which fits with observations that cOA$_4$ and cOA$_6$ are both activators of CARF nucleases. In contrast, cOA$_3$ and cOA$_5$ lack this symmetry and are unlikely to bind to CARF domains. They may be irrelevant in the context of signalling by type III systems, or alternatively may be activators for families of effector proteins that utilise a different domain for ligand binding.

To circumvent the difficulty in generating defined activators of CARF domain proteins, we have developed a facile technique that takes advantage of the specificity of the *E. coli* toxin MazF, which cleaves RNA 5' to an ACA sequence, generating a 3' cyclic phosphate group. Linear analogues of cOA can be generated with the same mass and charge as the cyclic equivalents in a cost effective and scalable manner and be radioactively labelled if required. This approach could prove particularly useful for smaller molecules such as OA$_4$, where phosphoramidite synthesis is problematic.

## Activation of cOA synthesis

Target RNA binding has been shown to activate both the HD nuclease and cOA synthesis activities of diverse type III systems. This is likely to result from conformational changes in the complex, and the Cas10 subunit in particular. A 5° rotation of the Cas10 subunit on target RNA binding has been observed for *T. thermophilus* Cmr (*Taylor et al., 2015*), which may reflect this activation. We observe that the activation of cOA synthesis is rapid, with activity detected within the first minute of incubation of Csm with target RNA transcripts. As for *S. thermophilus* Csm, activation requires a 3' extension to the target RNA that does not base-pair with the 5'-handle of the crRNA (*Figure 5* and [*Kazlauskiene et al., 2017*]). In our standard oligonucleotide substrates there is a 5 nt 3'- overhang, which fits the minimum requirement predicted from the biochemical and modelling study of the Siksnys group (*Kazlauskiene et al., 2016*). This short stretch of RNA presumably interacts with the Cas10 and/or the Csm4 subunit in such a way that a conformational change occurs in this region of the complex, resulting in the activation of both the HD nuclease and GGDD cyclase active sites, paving the way for DNA degradation and, through cOA signalling, the activation of HEPN family ribonucleases.

As observed for other type III systems, there is a high degree of tolerance for mismatches in target RNAs (*Pyenson et al., 2017*; *Manica et al., 2011*, *2013*; *Goldberg et al., 2018*), which all tend to be cleaved, and thus removed from the enzyme. By extending this analysis to examine cOA synthesis, we observe that 2 nt mismatches 5' to site B1 do activate the cyclase domain and would thus elicit a full anti-viral response. However, the target RNAs with a double mismatch at or 3' to site B1 (corresponding to positions within 4–5 nt of the 3' end of target RNA base-paired with the crRNA), whilst still cleaved, did not activate cOA production (*Figure 5*). This suggests an important role for the 3' end of the bound target RNA close to the Cas10 subunit where sequence requirements are more stringent. Presumably, destabilization of the RNA duplex in this region can lead to changes in the conformation of the single-stranded 3' end of the target RNA, resulting in failure to activate the cyclase domain. Viral targets that manage to mutate in this key area may thus have more chance of escaping from cOA-mediated HEPN nuclease activation, which is a key determinant of type III CRISPR immunity. In contrast, substantial blocks of mismatches in this general area do not prevent phage targeting by the *S. epidermidis* type III-A system (*Pyenson et al., 2017*). This could be due to differences between the activation of the cyclase and HD nuclease domains of Cas10, or to differences between type III-A and III-D systems and deserves further analysis.

The degree of specificity of the HEPN family nucleases against viral targets is still a matter for conjecture. The Marraffini group report that Csm6 selectively targets phage transcripts (*Jiang et al., 2016*), and the spatial coupling of the HD nuclease activity of Cas10 to actively transcribing viral RNA targets could ensure at least partial selectivity for foreign nucleic acids. Nonetheless, it seems likely that host nucleic acids would be caught in the cross-fire of this 'scorched earth' attack, leading to a shutdown in host gene expression or even cell death – which, as pointed out previously, could be a sensible option if viral infection is established in the cell (*Elmore et al., 2016*). Indeed, viral infection has been shown to result in dormancy and cell death *in S. islandicus*, which harbours a type III CRISPR system (*Bautista et al., 2015*).

## Switching off cOA synthesis

The activation of the HD nuclease and cyclase domains of Cas10 as a result of target RNA binding is not an irreversible event. Logically, it could be predicted that these domains are deactivated either when target RNA is cleaved, or when it dissociates from the complex, with a time lag possible in either circumstance. Backbone cleavage of RNA was the first activity detected in type III CRISPR systems (*Hale et al., 2009*), and appears to be ubiquitous. Nonetheless, this activity is not essential for type III-mediated immunity against phage infection in vivo (*Samai et al., 2015*). Terns and colleagues suggested that target RNA cleavage may represent a means for deactivation of the DNA nuclease activity of type III systems, as cleavage of the RNA transcript would cause the type III effector to 'drift away' (*Elmore et al., 2016*). Rather than spatial separation though, it seems more likely that target RNA cleavage and subsequent dissociation allows a collapse back to an inactive ground state of Cas10, switching off the highly potent HD nuclease and cOA signalling activities.

The Csm (III-A) complex from *S. thermophilus* and *Thermus thermophilus* also appear to cut target RNA first at the 3′ end nearest the Cas10 subunit (*Staals et al., 2014*; *Tamulaitis et al., 2014*), and cleavage by *S. thermophilus* Csm is rapid – of the order of 5 s under single turnover conditions (*Kazlauskiene et al., 2016*). *Thermatoga maritima* Cmr (III-B) cleaves target RNA with six nt spacing at four sites, with cleavage fastest at site 2, which is 11 nt from the 3′ end of the target RNA. Cleavage is rapid under single turnover conditions, reaching completion within 1 min for site 2 and 5 min for site 4, which is 23 nt from the 3′ end (*Estrella et al., 2016*). The rate of product release has not previously been quantified in any system. For *S. thermophilus* Csm, multiple turnover kinetic experiments suggested that product dissociation, whilst rate limiting, was moderately fast (k = 3 min$^{-1}$). However, activation of the DNase activity of the HD domain persisted over a much longer timescale, 30–60 min. The authors suggested that the activated state of *S. thermophilus* Csm persists after target RNA dissociation (*Kazlauskiene et al., 2016*), but an alternative is that there is heterogeneity in the population of Csm enzymes, with some remaining bound to RNA targets and hence preserved in an active state for much longer periods than others. For *T. maritima* Cmr, cleaved target RNA dissociation also correlated with the deactivation of the HD nuclease domain, although the rates were not measured (*Estrella et al., 2016*). Many of these uncertainties arise from the difficulty inherent in quantifying nuclease activity in these systems.

To delineate the roles of backbone cleavage and RNA dissociation in the control of cOA synthesis, we carried out a detailed kinetic analysis, quantifying the rates of target RNA cleavage and subsequent dissociation (*Figure 10*). We observed comparatively rapid cleavage at sites B1 (0.19 min$^{-1}$) and B2 (0.08 min$^{-1}$), nearest to the 3′ end of the target RNA and proximal to the Cas10 subunit, and much slower cleavage of more distal sites. The trend for faster cleavage at the 3′ end is clear from studies of other type III systems. The overall cleavage rate of a transcript RNA, which is a more physiologically relevant target, was very similar (*Figure 4*). The introduction of phosphorothioate linkages spanning sites B1 and B2 reduced but did not abolish target RNA cleavage (*Figure 8*), and we noted a highly significant increase in cOA production with these substrates, consistent with a link between target RNA cleavage and cOA synthesis. By trapping the dissociated RNA we could quantify the rate of product release directly for the first time for a type III system. We determined a rate of 0.07 min$^{-1}$ for dissociation of the 5′ end of the target RNA – a very close match to the rate of cleavage at site B2 in the centre of the RNA. Furthermore, the decrease in the rate of cOA synthesis by activated Csm over time also fitted an exponential with a rate of 0.05 min$^{-1}$, ten times faster than the rate of cleavage at site B3. Thus, it appears that full target RNA digestion may not be required for product release.

Furthermore, RNA dissociation is likely not rate limiting. Rather, the chemical step of RNA cleavage determines the rate of product release. As our assays were carried out at 70°C, close to the growth temperature of *S. solfataricus*, RNA duplexes of around 20 bp would be only marginally stable in the absence of stabilising protein interactions. This fits the situation in *S. islandicus*, where target RNA forming less than 20 bp with the crRNA was observed to dissociate (*Han et al., 2017a*). It is also consistent with the general observation for type III systems that target RNA cleavage products tend to accumulate under single turnover conditions rather than undergoing processive cleavage to the smallest possible products (*Staals et al., 2014*; *Tamulaitis et al., 2014*; *Hale et al., 2014*; *Han et al., 2017a*; *Zhang et al., 2016*). Finally, RNA cleavage and dissociation results in the simultaneous deactivation of the cyclase domain, presumably due to rapid conformational change to the

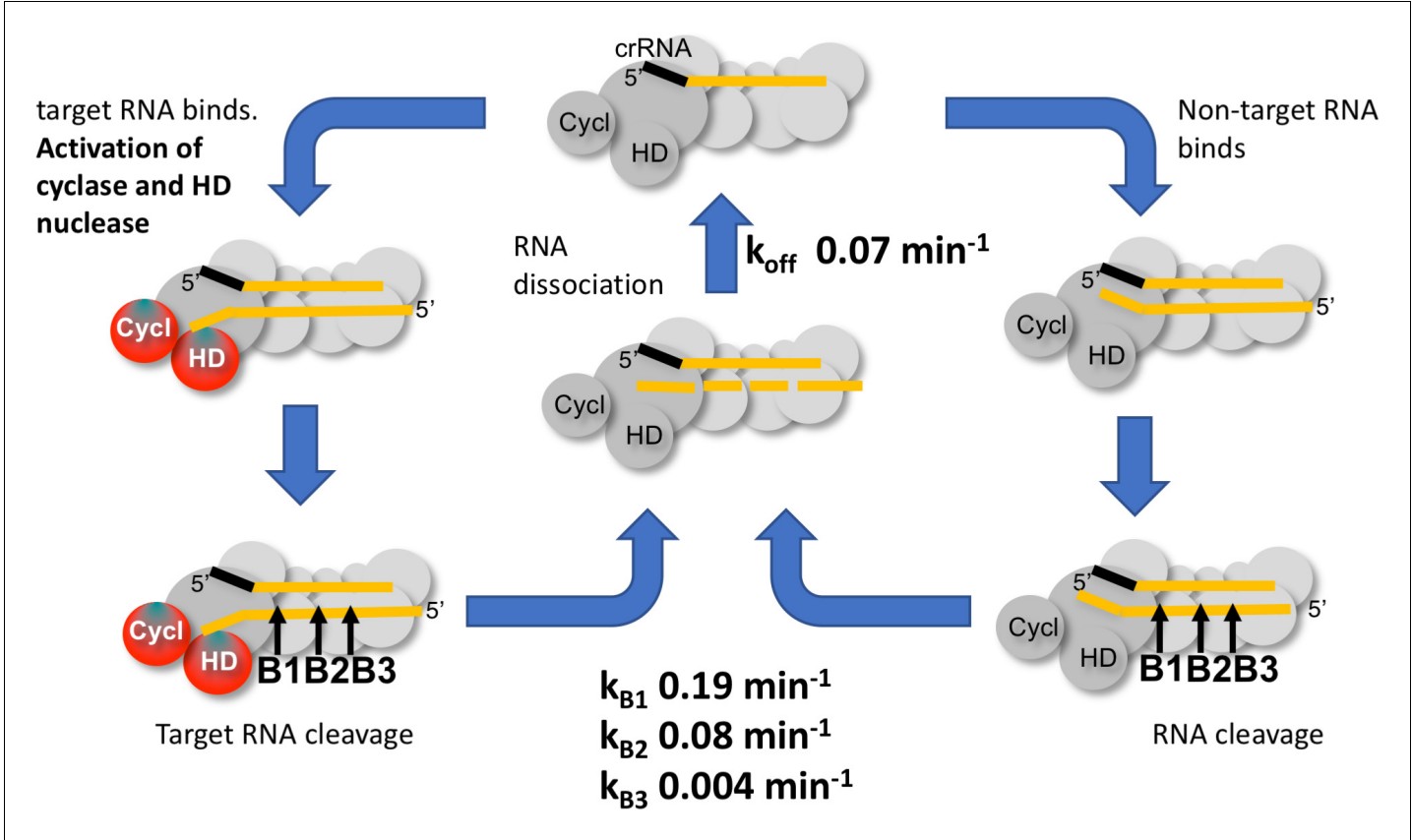

**Figure 10.** Kinetic scheme for activation and inactivation of cOA production. In its ground state, Csm adopts a conformation where the HD nuclease and cyclase domains are inactive. On binding cognate target RNA with a non-pairing 3' end, these domains are allosterically activated, stimulating DNA degradation and cOA synthesis. Backbone cleavage of the target RNA occurs most rapidly at site B1 nearest to the 3' end, with progressively slower rates of cleavage at sites B2 and B3. Cleaved RNA dissociation is observed with an off rate of 0.07 min$^{-1}$, suggesting it may be rate limited by cleavage at site B2 and does not require cleavage at site B3. The production of cOA is quickly switched off, consistent with a rapid return to the inactive ground state. Non-target RNAs proceed through the same processing steps, but do not activate the cyclase or HD nuclease domains.
DOI: https://doi.org/10.7554/eLife.36734.021

ground state. These observations are consistent with the hypothesis that target RNA cleavage (and therefore dissociation) in type III systems functions as an 'off switch' for the cyclase and HD nuclease active sites (*Estrella et al., 2016*). This, rather than the protective effect of direct degradation of viral transcripts, may be the primary role of target RNA cleavage. After all, a single target RNA may trigger synthesis of many molecules of cOA, which in turn would activate multiple HEPN ribonucleases for active RNA degradation – a potent amplification of antiviral defences. Under multiple turnover conditions that may be more reflective of a high viral load in the cell, the cyclase domain is activated for longer periods (*Figure 7*). RNA substrates that are not *bona fide* targets are still cleaved and thus cleared from the binding site, but do not activate cOA synthesis, allowing the Csm complex to seek other targets.

Several studies have reported that the HD nuclease activity of type III systems persists up to 1 hr, despite target RNA cleavage reaching completion within a few minutes (*Tamulaitis et al., 2014*; *Estrella et al., 2016*; *Kazlauskiene et al., 2016*). This raises the possibility that the HD nuclease and cyclase domains experience different half-lives in the activated state, with the former persisting for much longer. It could make sense to switch off the cyclase activity quickly, as the resultant cOA would provide a much more persistent activation of CARF-family nucleases and transcription factors. A systematic study of the temporal control of HD nuclease activity is required to resolve this question. Ultimately, the removal of the cOA second messenger by an as-yet unidentified

phosphodiesterase is likely to play an important part in the control of type III systems. This should be a priority for future study.

# Materials and methods

## Key resources table

| Reagent type (species) or resource | Designation | Source or reference | Identifiers | Additional information |
|---|---|---|---|---|
| Gene (*Sulfolobus solfataricus*) | Csm complex (eight subunits) | PMID: 24119402 | | virus expression construct |
| Gene (*Sulfolobus solfataricus*) | Csx1/Sso1389 | this paper | UniProtKB - Q97YD5 | plasmid expression construct |
| Gene (*Eschericia coli*) | MazEF | PMID: 22447587 | | plasmid expression construct |

## Csm complex expression and purification

The Csm (III-D) complex (wild-type, HD and GGDD variants) was expressed and purified from *S. solfataricus* as described previously (*Zhang et al., 2016*). This involves expression of a his-tagged subunit from an arabinose-inducible vector in *S. solfataricus*, followed by purification of the native complex with the tagged subunit incorporated. Variants such as the HD and GGDD targeted changes can thus be introduced on the tagged subunit. The resultant protein has a mixture of all the crRNAs loaded into the Csm complex, with abundant crRNAs such as A26 present at 1–2% of the total crRNA (*Rouillon et al., 2013*). A target RNA (target RNA A26 and derivatives) complementary to crRNA A26 is used in the assays described.

## Purification of Csx1 and Csx1 variant H345N

Sso1389 (encoding Csx1) was purchased as a synthetic gene from Integrated DNA Technologies, Coralville, IA, United States (IDT) and cloned into the pEHisTEV vector (*Liu and Naismith, 2009*). The inactive Csx1 variant H345N was generated using the QuikChange Site-Directed Mutagenesis kit as per manufacturer's instructions (Agilent technologies). The pEHisTEVCsx1 and pEHisTEVCsx1H345N constructs were transformed into C43 (DE3) *E. coli* cells. Protein expression was induced with 0.4 mM isopropyl-β-D-1-thiogalactoside (IPTG) at an $OD_{600}$ of ~0.6 and grown for 4 hr at 25°C. Cells were harvested and resuspended in lysis buffer containing 50 mM Tris-HCl pH 7.5, 0.5 M NaCl, 10 mM Imidazole and 10% glycerol, and lysed by sonicating six times 2 min on ice with 2 min rest intervals. Csx1 was purified with a 5 ml HisTrapFF column (GE Healthcare), washing with five column volumes (CV) of buffer containing 50 mM Tris-HCl pH 7.5, 0.5 M NaCl, 30 mM Imidazole and 10% glycerol, and eluting with a linear gradient of buffer containing 50 mM Tris-HCl pH 7.5, 0.5 M NaCl, 0.5 M imidazole and 10% glycerol across 15 CV. Size exclusion chromatography was used to further purify Csx1, eluting protein with buffer containing 20 mM Tris-HCl pH 7.5, 0.5 M NaCl, 1 mM EDTA and 1 mM DTT. Csx1 was concentrated using a centrifugal concentrator, aliquoted and frozen at −80°C.

## MazEF purification

The pET21c-mazEF-Fxa construct was a kind gift from Professor Masayori Inouye (Robert Wood Johnson Medical School, New Jersey). pET21c-mazEF-Fxa was transformed into BL21 (DE3) *E. coli* cells and expressed as previously published (*Park et al., 2012*), except cells were grown at 25°C for 4 hr after induction with 0.5 mM IPTG at $OD_{600}$ of ~0.6. Cells were harvested by centrifugation at 4000 x *g* at 4°C for 15 min, suspended in buffer A (10 mM Tris-HCl, pH 7.5 and 150 mM NaCl) and lysed by sonicating six times 2 min on ice with 2 min rest intervals. MazEF was purified with a 5 ml HisTrapFF column, washing with 20 CV buffer A and 20 CV 2% buffer B (10 mM Tris-HCl, pH 7.5, 150 mM NaCl, 1 M imidazole) prior to elution with a linear gradient of buffer B across 15 CV. Size exclusion chromatography (Sepharose200 26/600; GE Healthcare) was used to further purify MazEF, eluting protein with buffer A. MazEF was concentrated using a centrifugal concentrator, aliquoted and frozen at −80°C.

## RNA substrate preparation and kinetic analysis: short RNAs and transcripts

RNA oligonucleotides were ordered from IDT. Before assays, oligonucleotides were 5'-$^{32}$P-radiolabelled and purified by denaturing (7M urea) polyacrylamide (20%) gel electrophoresis with 1 x Tris-borate-EDTA (TBE) buffer, followed by band excision, gel extraction, ethanol precipitation, as described previously (*Rollie et al., 2015*). The RNA transcript substrate was generated using as template a *Hind*III linearised pUC19 vector containing a cloned DNA sequence (between *Bam*HI-*Hind*III) harbouring a target sequence (in bold) recognised by Csm crRNA A26 and a T7 phage polymerase promoter. Transcripts were obtained using the MEGAscript kit (Ambion by Life Technologies) according to the manufacturer instructions. $^{32}$P-α-ATP was added to the cold NTPs mix for further visualisation. Transcripts were finally purified using a G50 column (illustra Dye terminator removal kit-GE Healthcare) before use. For kinetic analysis, cleaved and uncleaved species were quantified using the Bio-Formats plugin (*Linkert et al., 2010*) of ImageJ (*Schneider et al., 2012*) as distributed in the Fiji package (*Schindelin et al., 2012*) and fitted to a single exponential curve using Kaleidagraph (Synergy Software), as described previously (*Sternberg et al., 2012*). For the transcript cleavage, a single experiment was quantified, but replicates were qualitatively similar. For the target A26 cleavage, the cleavage rate determined for site B1 could be a slight underestimate, as cleavage of B1 products at site B2 would decrease the B1 signal. This appears to be a minor effect however (see discussion).

The sequence of the DNA fragment used for the transcript generation, with the A26 target sequence in italics, is shown below:

5'GGATCCTAATACGACTCACTATAGGATCAGATCATATCAGCTACATCGACAGGGTATTATTTG
TTTGTTTCTTCTAAACTATAAGCTAGTTCTGGAGAGCATTAGCATGTAGAGGGTACAGTTTGGGTA
TTGCCGTTCTGGTCCTTATACGAAATGGAGATCGATTCTCGAGAGGGTCGTTGTTAAGAACGACG
TTGTTAGAAGTTGGGTATGGTGGAGATGGAAGCTT

## Csm target RNA cleavage and product release

The assay of target RNA cleavage by Csm was carried out as described previously (*Zhang et al., 2016*) unless specified. Briefly, the reaction was carried out under single turn-over conditions with 5 μM of CSM and 25 nM labelled RNA target in a buffer containing 20 mM MES pH 6.0, 100 mM NaCl, 0.1 mg/ml BSA, 2 mM MgCl$_2$ and 0.5 or 1 mM ATP as mentioned in the text. The reaction was incubated at 70°C for the specified times. Following the reaction, products were phenol-chloroform extracted and run in 50% formamide on a denaturing gel (20% acrylamide, 7M Urea, 1X TBE) to observe backbone cleavage. Alternatively, the same reaction products were chilled in a stop solution containing 2 mM EDTA and 10 μM of DNA trap before adding Ficoll and running on a native gel (12% acrylamide containing 50 mM of NaCl and 1X TBE). Following electrophoresis, gels were imaged by phosphorimaging as described previously (*Zhang et al., 2016*). For the experiments comparing unmodified and phosphorothioate-containing target RNA, the target RNA cleavage reactions were performed as described above in presence of α-$^{32}$P-ATP to visualise the formation of cyclic oligoadenylate.

Generation of cyclic oligoadenylate (cOA) activator was generated by incubating 120 μg *S. solfataricus* Csm complex with 0.5 mM ATP, 1 mM MgCl$_2$ and 100 nM target RNA in Csx1 buffer for 2 hr at 70°C. Reaction product was isolated by phenol-chloroform extraction followed by chloroform extraction and frozen at −20°C. For the experiments where the cOA production was observed over time, 2 nM of $^{32}$P-α-ATP was added (in addition to 1 or 0.5 mM of cold ATP) in the reaction to allow visualisation of the generated product on denaturing gel electrophoresis and phosphorimaging. In the case of the spiking experiment, $^{32}$P-α-ATP was added to a final concentration of 2 nM in the reaction (at 0, 20 or 40 min). The quantification of free $^{32}$P-α-ATP versus $^{32}$P-α-ATP incorporated in cOA allowed determination of a total ATP ratio used in cyclase reaction that is lower than 10% (50 μM) for the longest time points.

## Generation of active MazF and linear oligoadenylates

MazF was used to generate linear oligoadenylate. Active MazF was initially obtained by incubating 1 mg of MazEF with 0.1 units Factor X (Sigma-Aldrich) activated in FXa buffer containing 10 mM Tris-HCl pH 8.0 and 1 mM DTT for 3 hr at 37°C. MazEF digestion with bovine trypsin (Promega) (1600:1

ratio) was found to yield a similar level of active MazF when incubated in the same buffer for 15 min. Therefore, MazF was predominantly generated by trypsin digestion and used immediately. Linear oligoadenylate was generated by incubating MazF with 30 µM A3, A4, A5 or A6 RNA (*Table 1*) in FXa buffer for 2.5 hr at 37°C. Products were phenol-chloroform extracted, and then chloroform extracted and frozen at −20°C. Denaturing polyacrylamide gel electrophoresis (20% acrylamide, 7M urea and 1x TBE) followed by phosphorimaging of MazF-cleaved [32]P end-labelled A3, A4, A5 and A6 RNA was used to assess successful generation of intended linear OA.

## Csx1 activation assay

Csx1 and Csx1 H345N mutant were diluted in buffer containing 20 mM Tris-HCl pH 8.0, 0.5 M NaCl, 1 mM DTT and 1 mM EDTA. 250 nM of Csx1 dimer was incubated with 0.01% (v/v) cOA activator or 3 µM linear oligoadenylate (A3, A4, A5 or A6) and 50 nM [32]P-5'-labelled substrate RNA in Csx1 buffer containing 20 mM MES pH 5.5, 100 mM K-glutamate and 1 mM DTT. Two control reactions, both containing only 50 nM RNA and buffer were incubated at 4°C or 50°C, respectively. All other reactions were incubated at 50°C and quenched at 5, 15 or 30 min by the addition of a reaction volume equivalent of 100% formamide prior to freezing at −20°C. RNA cleavage by Csx1 was visualised by phosphorimaging following denaturing gel electrophoresis. Csx1 is stable and active at 70°C, but the choice of 50°C reduced background substrate RNA cleavage.

## Mass spectrometry

Liquid chromatography-high resolution mass spectrometry (LC-HRMS) analysis was performed on a Thermo Scientific Velos Pro instrument equipped with HESI source and Dionex UltiMate 3000 chromatography system. Compounds were separated on a Kinetex 2.6 µm EVO C18 column (2.1 × 100 mm, Phenomenex) using a linear gradient of 2–15% acetonitrile against 20 mM ammonium bicarbonate pH 8 (gradient start delay 5 min, gradient length 28 min) at a flow rate of 350 µl min$^{-1}$ and column temperature of 40°C. Data were acquired on the FT mass analyzer in negative ion mode with scan range $m/z$ 150–1500. Source voltage was set to 3.5 kV, capillary temperature was 350°C, and source heater temperature was 250°C.

## Acknowledgements

We thank Prof Masayori Inouye for the kind provision of the expression plasmid for *E. coli* MazEF, and Prof Virgis Siksnys for provision of a sample of cyclic oligoadenylate used for our preliminary screening. Thanks to the University of St Andrews Mass Spectrometry facility and Dr Clarissa Czekster for help with mass spectrometry. This work was supported by a grant from the Biotechnology and Biological Sciences Research Council (REF: BB/M000400/1 to MFW), and a Royal Society Challenge Grant (REF: CH160014 to MFW).

## Additional information

### Funding

| Funder | Grant reference number | Author |
| --- | --- | --- |
| Royal Society | Challenge grant CH160014 | Malcolm F White |
| Biotechnology and Biological Sciences Research Council | Project grant BB/M000400/1 | Malcolm F White |

The funders had no role in study design, data collection and interpretation, or the decision to submit the work for publication.

### Author contributions

Christophe Rouillon, Conceptualization, Formal analysis, Investigation, Methodology, Writing—original draft, Project administration, Writing—review and editing; Januka S Athukoralage, Sabine Grüschow, Investigation, Writing—original draft; Shirley Graham, Methodology, Writing—original draft;

Malcolm F White, Conceptualization, Formal analysis, Supervision, Funding acquisition, Writing—original draft, Project administration, Writing—review and editing

**Author ORCIDs**
Shirley Graham http://orcid.org/0000-0002-2608-3815
Malcolm F White http://orcid.org/0000-0003-1543-9342

**Decision letter and Author response**
Decision letter https://doi.org/10.7554/eLife.36734.023
Author response https://doi.org/10.7554/eLife.36734.024

## Additional files

### Data availability
All data generated or analysed during this study are included in the manuscript and supporting files.

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
