## [Decision Letter]

Thank you for submitting your article "Control of cyclic oligoadenylate synthesis in a type III CRISPR system" for consideration by *eLife*. Your article has been reviewed by three peer reviewers, and the evaluation has been overseen by Gisela Storz serving both as Reviewing and Senior Editor. The following individual involved in review of your submission has agreed to reveal his identity: Erik Sontheimer (Reviewer #3).

The reviewers have discussed the reviews with one another and the Reviewing Editor has drafted this decision to help you prepare a revised submission.

Summary:

In this article, the authors biochemically characterize the synthesis of cyclic oligoadenylate (cOA) by the Csm complex from a type III-D CRISPR-Cas system in *Sulfolobus solfataricus*. This activity has been studied previously in bacterial type III-A systems, but not in archaea, and not in type III-D. The authors purified the Csm complex, validated its nuclease activity, and showed that the cyclic oligoadenylates (cOAs) generated by the Cas10 subunit of Csm activate Csx1's ssRNA nuclease activity. As expected, their biochemical assays show that the production of cOA (mostly the tetramer) is induced upon target RNA binding to Csm. They further demonstrate that product release by the Csm complex is correlated with the inactivation of the cyclase activity of Cas10, possibly through conformational changes. Their creative method in using a bacterial toxin (MazF) to generate cOA species can be used by other researchers interested in studying cOA-related pathways. For the most part, the biochemistry is nicely done. There are relatively few unexpected findings here in light of what has been shown elsewhere, but nonetheless, work of this nature is important to advance our mechanistic view of type III CRISPR interference.

Essential revisions:

The study would be strengthened if the authors addressed the following points.

The first is experimental:

The authors posit a model in which RNA cleavage and subsequent product release deactivate cOA synthesis, and several lines of evidence (mainly correlated rates) support this. However, a key prediction is that *blocking* the cleavage of bound RNAs should slow product release and lead to nearly constitutive cOA synthesis. This key prediction was not tested even though it would add considerable strength to the conclusion. Such blockage could be achieved by mutating the Csm4 active site, or by using a target RNA that carries chemical modifications (perhaps 2'-O-methyl residues and/or phosphorothioates on both sides of the cleavage sites) that prevent hydrolysis. The latter would require the demonstration that such modified targets are still bound by the crRNA-loaded complex, but this should not be difficult (if the answer is yes).

The rest require rewording or further explanation:

- The authors state (including in the Abstract) that RNA shredding is likely to be important for cOA synthesis deactivation rather than direct antiviral defense. It is not clear why this needs to be framed as an either/or – why could both not be true?

- Figure 6: The rate constant measurement for k_B1_ and k_B2_ maybe problematic here. The authors need to provide a detailed description of how the rate constants were calculated. For example, B1 bands accumulated by cleavage at B1 position, but also diminish upon B2 cleavage.

- How did the authors reach the conclusion that "cleavage up to the centre of the target RNA (site B2) appears sufficient to allow product dissociation"? What is the identity of RNA 5' end + trap DNA in Figure 8A? Can you provide a B2RNA + trap DNA marker?

---

## [Author Response]

Essential revisions:The study would be strengthened if the authors addressed the following points.The first is experimental:The authors posit a model in which RNA cleavage and subsequent product release deactivate cOA synthesis, and several lines of evidence (mainly correlated rates) support this. However, a key prediction is that blocking the cleavage of bound RNAs should slow product release and lead to nearly constitutive cOA synthesis. This key prediction was not tested even though it would add considerable strength to the conclusion. Such blockage could be achieved by mutating the Csm4 active site, or by using a target RNA that carries chemical modifications (perhaps 2'-O-methyl residues and/or phosphorothioates on both sides of the cleavage sites) that prevent hydrolysis. The latter would require the demonstration that such modified targets are still bound by the crRNA-loaded complex, but this should not be difficult (if the answer is yes).

This is an excellent suggestion. We showed previously that we cannot inactivate the type III-D Csm system backbone cleavage by mutagenesis (Zhang et al., 2016). Accordingly, we designed a target RNA oligonucleotide with two sets of three phosphorothioate linkages spanning cleavage sites B1 and B2. This RNA target is cleaved significantly more slowly than the unmodified RNA and results in the synthesis of 4-fold higher levels of cOA. There is a particular reduction in cleavage at site B2, which fits with the idea that cleavage up to site B2 is important for product release. A new figure 8 and accompanying text have been added. We agree this strengthens the paper considerably.

The rest require rewording or further explanation:- The authors state (including in the Abstract) that RNA shredding is likely to be important for cOA synthesis deactivation rather than direct antiviral defense. It is not clear why this needs to be framed as an either/or – why could both not be true?

We still consider this to be the most likely scenario and feel it is important to raise this point, but we are happy to tone down our statements. In the Abstract, we altered this sentence to say that “RNA shredding […] *may thus be* a reflection”.

In the discussion, we rephrased the relevant section to read:

“These observations are consistent with the hypothesis that target RNA cleavage (and therefore dissociation) by type III systems functions as an “off switch” for the cyclase and HD nuclease active sites (Estrella, Kuo and Bailey, 2016). This, rather than the protective effect of direct degradation of viral transcripts, may be the primary role of target RNA cleavage.”

- Figure 6: The rate constant measurement for k_B1_ and k_B2_ maybe problematic here. The authors need to provide a detailed description of how the rate constants were calculated. For example, B1 bands accumulated by cleavage at B1 position, but also diminish upon B2 cleavage.

This is a good point. Although interconversion of B1 bands to B2 products is limited, we do see a slight decrease in B1 at later time points. The effect is limited, partly because not all the product is cleaved, but also seems to be a general property of these enzymes, as discussed in the Discussion in this section:

“It is also consistent with the general observation for type III systems that target RNA cleavage products tend to accumulate under single turnover conditions rather than undergoing processive cleavage to the smallest possible products (Staals et al., 2014; Tamulaitis et al., 2014; Hale et al., 2014; Liu and Naismith, 2009).”

Nevertheless, we agree with the referee that there could be a danger of underestimating the rate of B1 cleavage to some extent due to limited conversion to B2 products. We have therefore determined the rate constants for B1 and B2 cleavage from the early time points and have added the following sentence to the Materials and methods:

“For the transcript cleavage, a single experiment was quantified, but replicates were qualitatively similar. […] This appears to be a minor effect however (see Discussion).”

We have added further detail and appropriate citations for the methods used to calculate all rate constants:

“For kinetic analysis, cleaved and uncleaved species were quantified using the Bio-Formats plugin (Linkert et al., 2010) of ImageJ (Schneider, Rasband and Eliceiri, 2012) as distributed in the Fiji package (Schindelin et al., 2012) and fitted to a single exponential curve using Kaleidagraph (Synergy Software), as described previously (Sternberg, Haurwitz and Doudna, 2012).”

As we switched to this software package, which is optimised for the accurate quantification of phosphorimages, we recalculated all rates. This has resulted in small changes to the absolute rates as noted, but no changes to the overall interpretation of the data and conclusions drawn.

- How did the authors reach the conclusion that "cleavage up to the centre of the target RNA (site B2) appears sufficient to allow product dissociation"? What is the identity of RNA 5' end + trap DNA in Figure 8A? Can you provide a B2RNA + trap DNA marker?

As the referee points out, an important control was missing. We have provided an additional control by synthesising a synthetic RNA oligonucleotide (Target RNA B2 product) and demonstrating that, when hybridised to the DNA trapper, the heteroduplex runs at the same place in the gel as the main trapped product. This is now shown in Figure 9—figure supplement 1 (which was originally Figure 8).